# Comparison of acute appendicitis before and within lockdown period in COVID-19 era: A retrospective study from rural Nepal

**Suman Baral** [ORCID]*, **Raj Kumar Chhetri, Neeraj Thapa**

Department of Surgery, Lumbini Medical College, Tansen, Palpa, Nepal

* brylsuman.sur@gmail.com

**Data Availability Statement:** All relevant data are within the manuscript and its Supporting information files.

## Abstract

### Introduction

The world has been engulfed with the pandemic of the novel severe acute respiratory syndrome coronavirus 2 (SARS-CoV-2) which have created significant impact in the emergency surgical health delivery including acute appendicitis. The main aim of this study was to compare the demographic and clinical parameters between two cohorts before the onset of lockdown and within the pandemic.

### Methods

A retrospective analysis was performed between two groups A and B, who presented with acute appendicitis three months prior to and after initiation of lockdown on March 24 2020 respectively in one of the tertiary centers of Nepal. These two cohorts were compared in demographics, clinicopathological characteristics and surgical aspects of acute appendicitis.

### Results

There were 42 patients in group A and 50 patients in group B. Mean age of the patients was 31.32±17.18 years with male preponderance in group B (N = 29). Mean duration of pain increased significantly in group B [57.8±25.9(B) vs 42.3±25.0(A) hours, P = 0.004] along with mean duration of surgery [51.06±9.4(B) vs 45.27±11.8(A) minutes, P = 0.015]. There was significant decrease in post-operative hospital stay among group B patients [3.04±1.1 (B) vs 3.86±0.67(A) days, P = 0.0001]. Complicated cases increased in group B including appendicular perforation in 10 cases. Similarly, mean duration of presentation to hospital significantly increased in group B patients with perforation [69.6±21.01 vs 51.57±17.63 hours, P = 0.008].

### Conclusion

During the adversity of the current pandemic, increased number of cases of acute appendicitis can be dealt with surgery as the chances of late presentation and complexity of the lesion exists.

**Funding:** The author(s) received no specific funding for this work.

**Competing interests:** The authors have declared that no competing interests exist.

## Introduction

The whole world has been engulfed by the pandemic of the novel SARS-CoV-2 which was first seen in Late December in Wuhan, China [1]. The pathogenesis and clinical spectrum of the disease has been widely studied however the trends are changing day to day and new updates are being published. The current scenario has created a havoc throughout the world and most of the countries are trying to outweigh the deleterious effect of the contagion with strategies of social distancing and lock down to mitigate the serious outcome of the virus [2]. During the pandemic, the spectrum of the surgical emergencies are not supposed to decrease, however one can ascertain that the complexity of the lesion might be more severe owing to late presentation, pursuing home based treatment due to inaccessibility of transportation, fear of contracting the virus in the hospitals and denied treatment [3].

Acute appendicitis (AA) is one of the most common surgical emergencies worldwide with the life time risk of 7–8% [4]. Gold standard treatment for AA has been surgery for ages however recent trends have changed and guidelines were published regarding conservative treatment with antibiotic for the management of uncomplicated appendicitis. Jerusalem guidelines 2020 recommendation includes discussing Non-Operative Management (NOM) with antibiotics as a safe alternative to surgery in selected patients with uncomplicated AA and absence of appendicolith, advising of the possibility of failure and misdiagnosing complicated appendicitis [5]. It can be assumed that the number of emergency cases might decrease during pandemics along with complicated cases increasing due to late presentation to hospitals. This holds true in later case however, former might disagree in settings of low-income country, especially in tertiary centers which opened despite of the fear of COVID 19 while other private hospitals were closed. That means we tend to receive more emergency cases than previously experienced. Thus, the main aim of this study was to scrutinize the impact of coronavirus disease 2019 (COVID-19) on incidence, demography and patient characteristics in AA comparing with the equal time duration, before and after lockdown was initiated in the country.

## Methods

This was a retrospective observational study conducted at Department of Surgery, Lumbini Medical College and Teaching Hospital, Nepal. Ethical approval was taken from Institutional Review Board of Lumbini Medical College. (IRC No: IRC-LMC 21-D/020) All the patients admitted for surgery with the diagnosis of AA, 90 days prior to initiation of lockdown by Nepal government on March 24 2020 and 90 days post lockdown were included in the study. Electronic data base and discharge summaries of the patients from the concerned time duration were retrieved from July 1 to July 4 2020 and the required parameters and variables were filled up by the author themselves into the proforma designed. As the nature of the study was record based and retrospective, consent was waived by the ethical committee of the institute and was not obtained and data too were anonymized. Patients who were treated laparoscopically, open surgery or managed conservatively were the subjects of the study. Complications like appendicular perforation and abscess were also recorded. The subject groups were categorized as Group A, who presented within 90 days prior to lockdown and Group B- within 90 days post initiation of the lockdown. Patient demography, clinicopathological variables along with intraoperative findings were determined. IBM SPSS Statistics ® 16.0 was used for statistical analysis. Descriptive variables were assessed as mean with standard deviation (SD), categorical variables between two groups were compared using Chi-squared test or Fisher's exact test and continuous variables were tabulated using Student's T test whichever applicable. Comparison between presence of fecalith and associated perforation along with time duration of abdominal pain were also looked for to find any significant association due to delayed

presentation to the hospital. It was assumed that after lockdown initiation, there might be significant changes in clinical parameters and outcomes due to delayed presentation to the hospital. P value of less than 0.05 was considered significant.

## Results

There were total 42 patients in group A and 50 patients in group B. Leukocytosis more than 10000/mm$^3$ was seen in 20 patients in group A while 39 patients showed leukocytosis in group B which was statistically significant. (P = 0.002) Duration of pain showed statistically significant difference between two groups. (P = 0.004) Mean overall Alvarado score was 6.54 ± 1.9 while it was higher in group B patients. {7.22±1.4 in group B vs 5.74±2.1 in group A, P value-0.0001}. Time duration of surgery increased in group B in comparison to group A with statistical significance. Among total patients; 2 cases had normal appendix, 63 had uncomplicated appendicitis while 19 had complicated appendicitis which included 7 patients in group A and 12 patients in group B. Post-operative hospital stay duration tended to decrease in patients operated after lockdown was initiated. (3.86 vs 3.04 days, P value = 0.0001) Total patients who presented with perforation were 16 in number which included 10 patients in COVID era and six were before lockdown. Perforation rate increased by 5.8%. Twelve patients in Group B had complicated appendicitis while there were 7 patients in group A with complicated appendicitis. There was no mortality (Table 1).

Group A (N = 37) patients had six appendicular perforations with one fecalith. Patients without fecalith were two cases of eight- and 10-year-old boys who presented after three days and seven days of pain abdomen respectively. The third one was 52 years male who was diabetic under medication. Other two patients had no other co-morbidities. Group B (N = 47) patients had 10 perforations which included six fecaliths as the source of perforation. (P = 0.0001) Three out of 4 cases without fecalith had Diabetes Mellitus 2 (DM-2) while the fourth patient was the seven-year-old boy who presented after 72 hours of onset of abdominal pain. (Table 2).

The mean duration of pain abdomen for perforated cases was 69.6±21.0 hours while it was 51.57±17.63 hours for non-perforated cases in group B patients (P = 0.008) Similarly, Group A

**Table 1. A comparison between two groups among demography and clinical parameters of patients presenting with AA.**

| Variable | | Total (N = 92) | Group A (N = 42) | Group B(N = 50) | P value |
|---|---|---|---|---|---|
| Age (Years) ± SD | | 31.32±17.18 | 30.17±16.12 | 32.28±18.13 | 0.56 |
| Sex | Male | 49 | 20 | 29 | 0.32 |
| | Female | 43 | 22 | 21 | |
| Leukocytosis>10,000/mm$^3$ | | 59 | 20 | 39 | 0.002 |
| Duration of pain (Hours)±SD | | 50±28.5 | 42.3±25.0 | 57.8±25.9 | 0.004 |
| Delayed presentation (>72 hours) | | 8 | 3 | 5 | 0.62 |
| **Treatment Modality** | | | | | |
| Conservative | | 8 | 5 | 3 | 0.001 |
| Laparoscopic | | 9 | 9 | 0 | |
| Open | | 75 | 28 | 47 | |
| Mean time duration of surgery (minutes) ± SD | | 48.5±10.8 | 45.27±11.8 | 51.06±9.4 | 0.015 |
| Mean Post-Operative Hospital Stay (days) ± SD | | 3.51±1.16 | 3.86 ±0.67 | 3.04±1.1 | 0.0001 |
| Perforation | | 16 | 6 | 10 | 0.55 |
| Perforation rate (%) | | 17.39% | 14.2% | 20% | |

**Table 2. Showing the relationship between presence of fecalith and perforation between two groups.**

| | | Fecalith | | P value |
|---|---|---|---|---|
| | Perforation | Absent | Present | |
| Group A(N = 37) | Absent | 31 | 0 | |
| | Present | 5 | 1 | 0.16* |
| Group B(N = 47) | Absent | 36 | 1 | |
| | Present | 4 | 6 | 0.0001* |

*Fishers exact test.

patients showed increased tendency in mean duration of abdominal pain before presentation to hospital however, there was no any statistical significance. (Table 3).

## Discussion

The study we conducted at a rural part of the country Nepal, came up with various findings especially associated with COVID-19 era comparing the findings with non COVID times. We found few studies related to appendicitis during this pandemic and most of them arrived to a conclusion of seeing a lesser number of patients coming to emergencies amidst the present scenario [6, 7]. In contrary, we could see a greater number of patients surfacing to the emergencies and getting operated in comparison to the same time frame before this contagious disaster at our clinical setting. The valid reason for this disparity, though small in number, could be due to closure of private hospitals around the area after the lockdown was initiated, from where the patient turns up. Also, the pooling occurred at our institute as this serves as one of the tertiary centers in the region. The duration of pain abdomen before presentation to the hospital significantly increased between two groups, latter showing 57.8±25.9 hours (P = 0.004). The scenario of delayed presentation to health care centers has been since ages in developing countries due to unaffordability issues, difficult geographical topography and lack of adequate transportation during normal days which itself was compromised. These obligations were further accentuated by the blooming contagion that led to harrowing consequences, which can be anticipated during these times of strict immobility. Patients were being confined to homes and taking home based treatments in the fear and anxiety of contracting virus from health care personnel and hospitals [8]. This further aggravated the diseased status of the patient arching to complexity which can be exemplified by the increased rates of perforation and complicated cases in our setting. Appendicular perforation is one of the dreaded complications of late presentation to the hospital which increases morbidity and mortality in comparison to non-complicated appendicitis. Studies have shown the perforation rates ranging from 16% to 40% [5]. Our study showed almost similar rates of perforation though there was slight increase in perforation rates in group B by six percent comparing to group A patients (14.2% vs 20%) though statistically insignificant, while total complicated cases increased by around

**Table 3. Showing the association between perforation and mean duration of abdominal pain.**

| Patient characteristic | Perforation | Mean duration of abdominal pain (hours) | P value |
|---|---|---|---|
| Group A | No | 40.39±25.43 | 0.18 |
| | Yes | 56.0±29.06 | |
| Group B | No | 51.57±17.63 | 0.008 |
| | Yes | 69.6±21.01 | |

7% (16.67% vs 24%). Snapiri et al. [9] showed that total complication rates of 22% during COVID times were similar to the study by Tankel et al. [6] Overall prevalence of fecalith was 9.5% in our study in which 16.67% of perforations in group A and 60% of perforations in group B were fecalith induced. The prevalence rate was somehow similar to the study from West Indies by Ramdass et al. [10] where fecaliths were present in 13.6% of the appendectomy specimens.

With the advent of first minimal invasive surgery as laparoscopic appendectomy in 1983 by Semm [11], the treatment of this most common surgical emergencies has seen an immense shift of treatment procedure from open approach to laparoscopy, however open approaches have not been abandoned in low middle-income countries like Nepal. Instead, we could see a surge of open surgeries in comparison to laparoscopy especially in the rural parts of the country. Potential advantages of early return to work, minimal hospital stay, minimized post-operative surgical site infections have been documented in literatures as the beneficial outcomes in comparison to open appendectomies [12], however the cost factor plays a critical role in low economies where surgical health is still primitive and laparoscopic advancements in rural part of the country is still primordial. Also, the surgical choice is to be decided by the patient and still the traditional open approaches are considered by them. Lower number of laparoscopies before COVID-19 and no any laparoscopic appendectomy during lockdown was evident in our clinical practice. The main reason behind this low digit was due to patient reluctance for laparoscopic surgery and cost factors which surpass open appendectomy. However, during lockdown, laparoscopy was completely abandoned at our setting as various controversies existed regarding aerosol generating procedures and safety of the health care workers too were inconclusive [13].

The mean operative time duration increased significantly between two groups, group B showing increased mean duration in comparison to group A. This could be due to extra precautions taken by the operating surgeons, virtually limiting chances of prick injuries while trying the best to limit complications to occur. Similarly, operating while wearing Personal Protective Equipment (PPE) with a foggy visibility along with complicated appendicitis encountered mandated extra cautiousness to take into account. Various literatures have shown that the duration of surgery is longer in laparoscopy group than open techniques [14, 15], however our experience suggests no any significance regarding the technique of the surgery as both the laparoscopic and open appendectomies took almost similar time duration. Considering the teaching institute, most of the open appendectomies were performed by the surgical residents under supervision while minimal laparoscopies performed were done by the experienced consultants which also may be one of the factors of increasing time duration of surgery in group B where no laparoscopy was considered. Tankel et al. [6] in his publication accounted the mean duration of surgery for 47.2 ± 28.9 minutes which almost corroborates our timing of 48.5 ± 10.8 minutes. Around two to six percent of cases with AA present with appendiceal mass which mainly includes inflammatory phlegmon or abscess [16]. Overall rate of appendicular lump was 7.6% in our study. Only three cases were managed conservatively in group B which included appendicular lump in two cases. Duration of hospital stay tend to decrease in group B patients in our study with statistical significance in comparison to group A. Delayed presentation to hospital along with complicated appendicitis like perforation seem to have prolonged hospital stay in various literatures [17, 18], however, this was not evident in our context especially during the time of pandemic. This could be due to patients willing to get discharged early once operated if feasible, minimizing the risk of protraction of the virus from other patients who have been hospitalized. Also, this practice allowed the rapid turnover of the patients allowing void of the beds that may be required in times of crises if surge of the COVID cases were to be seen in the forthcoming days.

Our study depicted that the surgical approach that was mandated at our institute for long before the pandemic ensued, is still being followed. The treatment strategy for the cases of AA was solely based upon the clinical judgement of the surgeon whether to operate or not, maximizing the use of protective gears with minimum use of man power in operating room. The principle of treating the primary cause rather than the symptoms of the disease was not violated keeping in mind the burden of the present contagion scenario which seems quiet less in developing countries like Nepal in comparison to the other parts of the world. Similarly, there might be some obligations for proceeding with conservative approaches with antibiotics alone in the setting of a low-income country where radiological investigations like contrast enhanced computed tomography (CECT) might not be feasible or available in order to diagnose non complicated appendicitis and rule out complicated cases. What our experience suggests is the cost factor if tabulated while performing abdominal CT along with fetching antibiotics almost completes the surgery. The financial aspects also need to be considered while working on low resource settings like ours where the needy ones are striving for surgical health and financial burden needs to be mitigated providing the definite care in a low budget scenario.

The lesser time duration and lower number of cases are the limitations of the present study. COVID- 19 is a new disease and we could not find similar studies from the region to compare and discuss the findings. We believe as the duration of the study is 3 months pre and post COVID era within lockdown and with the limited number of cases, the statistics may not be fully relied upon as this may be misleading as exemplified in the perforation rates between two groups. Also, this is a single center analysis of the patients with a smaller sample size which might not cover all the demographic and clinical aspects of the cohorts. The novice nature of this contagion which seems to involve the gastrointestinal system might even affect the clinical course of appendicitis which is yet to be elucidated, in which case the number of samples might increase along the parameters. None of the patients had testing for coronavirus as the tests were limited, costly and reserved for symptomatic or suspected cases. Still, we are experiencing positive cases without symptoms while tracing contacts, there would have been cohorts with positivity of the virus without symptoms if tests were implemented which would have changed the treatment modality for suspected uncomplicated cases.

## Conclusion

During the adversity of COVID-19, number of cases of AA, duration of presentation to hospital and complicated cases along with the perforation rate tend to increase in comparison to the cohort before the pandemic. Appendectomy should be the mainstay of treatment as the conservative approach in the fear of the pandemic might not be cost effective in areas of low-income countries.

## Supporting information

**S1 File.**
(SAV)

## Acknowledgments

The current manuscript has been uploaded as a pre-print in the Research Square and the assigned DOI number is 10.21203/rs.3.rs-47510/v2. However, no peer review has been done and this does not constitute the dual publication.

## Author Contributions

**Conceptualization:** Suman Baral, Raj Kumar Chhetri, Neeraj Thapa.

**Data curation:** Suman Baral.

**Investigation:** Suman Baral.

**Methodology:** Suman Baral.

**Software:** Suman Baral.

**Supervision:** Neeraj Thapa.

**Validation:** Suman Baral.

**Writing – original draft:** Suman Baral.

**Writing – review & editing:** Suman Baral, Raj Kumar Chhetri.

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
