## [Decision Letter · Decision Letter 0]

14 Oct 2020

PONE-D-20-24725

Comparison of acute appendicitis before and within COVID-19 era: A retrospective study from rural Nepal

PLOS ONE

Dear Dr. Baral,

Thank you for submitting your manuscript to PLOS ONE. After careful consideration, we feel that it has merit but does not fully meet PLOS ONE’s publication criteria as it currently stands. Therefore, we invite you to submit a revised version of the manuscript that addresses the points raised during the review process.

Please revise accordingly.

We look forward to receiving your revised manuscript.

Kind regards,

Academic Editor

PLOS ONE

Journal Requirements:

3. Please include the date(s) on which you accessed the databases or records to obtain the data used in your study.

4. We noted in your submission details that a portion of your manuscript may have been presented or published elsewhere.

"The manuscript has been uploaded as pre-print in Research Square."

Please clarify whether this publication was peer-reviewed and formally published. If this work was previously peer-reviewed and published, in the cover letter please provide the reason that this work does not constitute dual publication and should be included in the current manuscript.

Reviewers' comments:

Reviewer's Responses to Questions

**Comments to the Author**

1. Is the manuscript technically sound, and do the data support the conclusions?

Reviewer #1: Partly

Reviewer #2: No

2. Has the statistical analysis been performed appropriately and rigorously? 

Reviewer #1: I Don't Know

Reviewer #2: No

3. Have the authors made all data underlying the findings in their manuscript fully available?

Reviewer #1: Yes

Reviewer #2: No

4. Is the manuscript presented in an intelligible fashion and written in standard English?

Reviewer #1: Yes

Reviewer #2: Yes

5. Review Comments to the Author

Reviewer #1: The framework of the study is well assembled. The purpose is mentioned in the introduction and well followed with concrete data in the text thus leading to a direct, simple and interesting conclusion.

Although the sample sizes are relatively small, one must take into consideration the size of the region and population where the study was done. The conclusions are what we were expecting, that patients tended to increase the time until they arrive at the hospital during the lock-down.

The are a few language mistakes, the article should be re-read by a native speaker.

Table nr. 1 if possible should be redone, and the number of rows decreased as only the essential variables should be compared. The others could be added in plain text.

Congrats to the authors for their preoccupation.

Reviewer #2: The authors attempt to elucidate the topic of "Comparison of acute appendicitis before and within COVID-19 era", but not all of the patients had been tested for coronavirus in such a small sample size. In this situation, it should be hard to make any conclusion on this point as your title.

6. PLOS authors have the option to publish the peer review history of their article (what does this mean?). If published, this will include your full peer review and any attached files.

Reviewer #1: No

Reviewer #2: No

---

## [Author Response · Author response to Decision Letter 0]

23 Nov 2020

Editorial comments

1.Please ensure that your manuscript meets PLOS ONE's style requirements, including those for file naming.

# Response: Ensured: Title page and manuscript page changed as suggested

 # Response: As the nature of the study was record based and retrospective, consent was waived by the ethical committee of the institute and was not obtained and data too were anonymized (Sentence added: 8-10th line- Methods)

3. Please include the date(s) on which you accessed the databases or records to obtain the data used in your study.

# Response: from July 1 to July 4 2020 (Phrase added on Line 7- Methods)

4. We noted in your submission details that a portion of your manuscript may have been presented or published elsewhere.

"The manuscript has been uploaded as pre-print in Research Square."

Please clarify whether this publication was peer-reviewed and formally published. If this work was previously peer-reviewed and published, in the cover letter please provide the reason that this work does not constitute dual publication and should be included in the current manuscript.

# Response: The current manuscript has been uploaded as a pre-print in the Research Square and the assigned DOI number is 10.21203/rs.3.rs-47510/v2. However, no peer review has been done and this does not constitute the dual publication. (added in Acknowledgement) 

Reviewers Comments

# Reviewer 1

1. Table nr. 1 if possible, should be redone, and the number of rows decreased as only the essential variables should be compared. The others could be added in plain text.

Response# Modified as suggested!

# Reviewer 2

1. The authors attempt to elucidate the topic of "Comparison of acute appendicitis before and within COVID-19 era", but not all of the patients had been tested for coronavirus in such a small sample size. In this situation, it should be hard to make any conclusion on this point as your title.

Response# Title modification done:

Old title: Comparison of acute appendicitis before and within COVID-19 era: A retrospective study from rural Nepal

New title: Comparison of acute appendicitis before and within lockdown period in COVID-19 era: A retrospective study from rural Nepal

# Reviewer 3

1. In Group A, more patient received open appendectomy rather than laparoscopic appendectomy (75 VS 9). In Group B, no patient underwent laparoscopic operation (47 VS 0). Since the mainstream of appendectomy around the world is through endoscope due to lower/ similar complication rate and shorter hospital stay, why did few patients receive laparoscopic surgery, particularly in Group B? Notably, it’s seems to be more controllable for infection during endoscopic surgery rather than open surgery. This topic shall be addressed.

# Comment: With the advent of first minimal invasive surgery as laparoscopic appendectomy in 1983 by Semm, the treatment of this most common surgical emergencies has seen an immense shift of treatment procedure from open approach to laparoscopy, however open approaches have not been abandoned in low middle-income countries like Nepal. Instead, we could see a surge of open surgeries in comparison to laparoscopy especially in the rural parts of the country. Potential advantages of early return to work, minimal hospital stay, minimized post-operative surgical site infections have been documented in literatures as the beneficial outcomes in comparison to open appendectomies, however the cost factor plays a critical role in low economies where surgical health is still primitive and laparoscopic advancements in rural part of the country is still to perish. Also, the surgical choice is to be decided by the patient and still the traditional open approaches are considered by them. Lower number of laparoscopies before COVID-19 and no any laparoscopic appendectomy during lockdown was evident in our clinical practice. The main reason behind this low digit was due to patient reluctance for laparoscopic surgery and cost factors which surpasses open appendectomy. However, during lockdown, laparoscopy was completely abandoned at our setting as various controversies existed regarding aerosol generating procedures and safety of the heath care workers too were inconclusive.: Line 30- 44 added in Discussion

2. According to comment 1, the duration of surgery, post OP hospital stay, or perforation rate shall be discussed in open surgery or laparoscopic operation in Group A and Group B, respectively. That is, in addition to infection control or additional preparation, did the mean time duration of surgery in Group B longer than the time in Group A may result from the type of operation? The author shall clarify this issue.

# Response: Various literatures have shown that the duration of surgery is longer in laparoscopy group than open techniques, however our experience suggests no any significance regarding the technique of the surgery as both the laparoscopic and open appendectomies took almost similar time duration. Considering the teaching institute, most of the open appendectomies were performed by the surgical residents under supervision while minimal laparoscopies performed were done by the experienced consultants which also may be one of the factors of increasing time duration of surgery in group B where no laparoscopy was considered. Line 50-56 added in discussion 

3. In Table 2 describing the relationship between the presence of fecalith and perforation, why the group A is only 37 people and group B is 47 people? The case number is different from cases included? What is the missing cases?

# Response: Missing cases (N= 8) were managed conservatively. They were not operated.

4. The author concluded the number of cases of acute appendicitis increase after pandemic of COVID-19. However, the author did not offered a substantial evidence of the growing number of cases. Similarly, the author concluded that the ratio of complicated case (perforation) increase after pandemic of COVID-19. However, according to the Table 1, there is no statistically significant difference between the differences of perforation rate in two group.

# Response:

1. In contrary, we could see a greater number of patients surfacing to the emergencies and getting operated in comparison to the same time frame before this contagious disaster. The valid reason for this disparity, though small in number, could be due to closure of private hospitals around the area after the lockdown was initiated, from where the patient turns up and the pooling occurred at our institute as this serves as one of the tertiary centers in the region. (Line 4-9, Discussion)

2. The lesser time duration and lower number of cases are the limitations of the present study. COVID- 19 is a new disease and we could not find similar studies from the region to compare and discuss the findings. We believe as the duration of the study is 3 months pre and post COVID era within lockdown and with the limited number of cases, the statistics may not be fully relied upon as this may be misleading as exemplified in the perforation rates between two groups. Also, this is a single center analysis of the patients with a smaller sample size which might not cover all the demographic and clinical aspects of the cohorts. Line 83-90, Discussion)

---

## [Decision Letter · Decision Letter 1]

30 Nov 2020

PONE-D-20-24725R1

Comparison of acute appendicitis before and within lockdown period in COVID-19 era: A retrospective study from rural Nepal

PLOS ONE

Dear Dr. Baral,

Thank you for submitting your manuscript to PLOS ONE. After careful consideration, we feel that it has merit but does not fully meet PLOS ONE’s publication criteria as it currently stands. Therefore, we invite you to submit a revised version of the manuscript that addresses the points raised during the review process.

Please revise accordingly.

We look forward to receiving your revised manuscript.

Kind regards,

Academic Editor

PLOS ONE

Reviewers' comments:

Reviewer's Responses to Questions

**Comments to the Author**

1. If the authors have adequately addressed your comments raised in a previous round of review and you feel that this manuscript is now acceptable for publication, you may indicate that here to bypass the “Comments to the Author” section, enter your conflict of interest statement in the “Confidential to Editor” section, and submit your "Accept" recommendation.

Reviewer #1: (No Response)

Reviewer #4: (No Response)

2. Is the manuscript technically sound, and do the data support the conclusions?

Reviewer #1: Partly

Reviewer #4: Partly

3. Has the statistical analysis been performed appropriately and rigorously? 

Reviewer #1: Yes

Reviewer #4: Yes

4. Have the authors made all data underlying the findings in their manuscript fully available?

Reviewer #1: Yes

Reviewer #4: Yes

5. Is the manuscript presented in an intelligible fashion and written in standard English?

Reviewer #1: No

Reviewer #4: Yes

6. Review Comments to the Author

Reviewer #1: There are multiple english mistakes in the text. It needs to be read by a native speaker and corrected. I have highlighted some of them in the attached document. Also the article contains very long sentences especially in the discussion segment, some of them have 4-5 rows and very hard to follow. Please change accordingly.

Some mentions lack citations. I have highlighted them in the text.

In the results:

The authors mention: Leukocytosis more than 10000/mm3 was seen in 20 patients in group A while 39 patients showed increased WBC (White

Blood Cells) count in group B which was statistically significant.

The authors need to be more careful with words: leukocytosis and while blood cells are two different things. Please change accordingly.

Reviewer #4: The article does highlight the impact of Covid-19 on the management of the most common surgical emergency, that is AA. The discussion goes on and on and there are a few things that have been repeated in the intro as well as the discussion. As far as the sample size is concerned, i believe it is too small a sample to reach to a definite conclusion.

The results say that there was increased pain and increased number of cases with perforation in Group B. Moreover the duration of surgery was increased in Group B. But these points are not addressed when the results say that the duration of hospital stay was decreased in Group B and that too with a high statistical difference of almost a day.

Is there any association of presence of fecalith with Covid-19 or it is just a mere coincidence?

7. PLOS authors have the option to publish the peer review history of their article (what does this mean?). If published, this will include your full peer review and any attached files.

Reviewer #1: No

Reviewer #4: **Yes: **Waqas Ahmed

---

## [Author Response · Author response to Decision Letter 1]

19 Dec 2020

19/12/2020

The Editor in Chief

PLOS One

Re: Resubmission of the manuscript along with rebuttal letter as per suggestions 

Sir

Please kindly find the below mentioned response for the suggestions as provided by the esteemed editors and reviewers. I have tried my best to incorporate the shortcomings as suggested and meet upon the expectations. Please feel free to contact me for any corrections needed.

Sincerely

Suman Baral

(P.S: The responses go as below) 

Reviewer 1 Comments

1. In the results:

The authors mention: Leukocytosis more than 10000/mm3 was seen in 20 patients in group A while 39 patients showed increased WBC (White

Blood Cells) count in group B which was statistically significant.

The authors need to be more careful with words: leukocytosis and while blood cells are two different things. Please change accordingly.

# Response: Sentence changed 

Leukocytosis more than 10000/mm3 was seen in 20 patients in group A while 39 patients showed leukocytosis in group B which was statistically significant. (P= 0.002)

2. There are multiple english mistakes in the text. It needs to be read by a native speaker and corrected. I have highlighted some of them in the attached document. Also the article contains very long sentences especially in the discussion segment, some of them have 4-5 rows and very hard to follow. Please change accordingly.

Some mentions lack citations. I have highlighted them in the text.

# Response: As suggested- Changes made.

Reviewer 4 Comments

1. The article does highlight the impact of Covid-19 on the management of the most common surgical emergency, that is AA. The discussion goes on and on and there are a few things that have been repeated in the intro as well as the discussion. As far as the sample size is concerned, i believe it is too small a sample to reach to a definite conclusion.

# Response: Regarding the sample size, I too believe the number of cases is small, but considering the time duration and being a single center comparison study, the number of cases seems logical and this has been the limitation of our study.

2. The results say that there was increased pain and increased number of cases with perforation in Group B. Moreover, the duration of surgery was increased in Group B. But these points are not addressed when the results say that the duration of hospital stay was decreased in Group B and that too with a high statistical difference of almost a day.

# Response: Duration of hospital stay tend to decrease in group B patients in our study with statistical significance in comparison to group A. Delayed presentation to hospital along with complicated appendicitis like perforation seem to have prolonged hospital stay in various literatures,[17, 18] however, this was not evident in our context especially during the time of pandemic. This could be due to patients willing to get discharged early once operated if feasible, minimizing the risk of protraction of the virus from other patients who have been hospitalized. Also, this practice allowed the rapid turnover of the patients allowing void of the beds that may be required in times of crises if surge of the COVID cases were to be seen in the forthcoming days. (Line added- Discussion- Paragraph 3 – Line 17-25)

3. Is there any association of presence of fecalith with Covid-19 or it is just a mere coincidence?

# Coincidence may be! No any studies were found in the literature regarding this matter.

---

## [Decision Letter · Decision Letter 2]

23 Dec 2020

Comparison of acute appendicitis before and within lockdown period in COVID-19 era: A retrospective study from rural Nepal

PONE-D-20-24725R2

Dear Dr. Baral,

We’re pleased to inform you that your manuscript has been judged scientifically suitable for publication and will be formally accepted for publication once it meets all outstanding technical requirements.

Kind regards,

Academic Editor

PLOS ONE

Additional Editor Comments (optional):

Reviewers' comments:

Reviewer's Responses to Questions

**Comments to the Author**

1. If the authors have adequately addressed your comments raised in a previous round of review and you feel that this manuscript is now acceptable for publication, you may indicate that here to bypass the “Comments to the Author” section, enter your conflict of interest statement in the “Confidential to Editor” section, and submit your "Accept" recommendation.

Reviewer #1: All comments have been addressed

Reviewer #5: All comments have been addressed

2. Is the manuscript technically sound, and do the data support the conclusions?

Reviewer #1: Yes

Reviewer #5: Yes

3. Has the statistical analysis been performed appropriately and rigorously? 

Reviewer #1: Yes

Reviewer #5: Yes

4. Have the authors made all data underlying the findings in their manuscript fully available?

Reviewer #1: Yes

Reviewer #5: Yes

5. Is the manuscript presented in an intelligible fashion and written in standard English?

Reviewer #1: Yes

Reviewer #5: Yes

6. Review Comments to the Author

Reviewer #1: The authors have resolved the raised issues with the article. From my own personal point of view the article is fit for publication.

Reviewer #5: I have been entrusted the task of reviewing the revised manuscript after valuable suggestions of my reviewing colleagues. I found the revised manuscript appropriate now. The language, flow of text and conclusions are in much better format.

7. PLOS authors have the option to publish the peer review history of their article (what does this mean?). If published, this will include your full peer review and any attached files.

Reviewer #1: No

Reviewer #5: No

---

## [Editor Report · Acceptance letter]

28 Dec 2020

PONE-D-20-24725R2 

Comparison of acute appendicitis before and within lockdown period in COVID-19 era: A retrospective study from rural Nepal 

Dear Dr. Baral:

I'm pleased to inform you that your manuscript has been deemed suitable for publication in PLOS ONE. Congratulations! Your manuscript is now with our production department. 

Kind regards, 

on behalf of

Dr. Robert Jeenchen Chen 

Academic Editor

PLOS ONE